

**A fifteen year record of CO emissions constrained by MOPITT CO observations**

Zhe Jiang[1,2], John R. Worden[1], Helen Worden[2], Merritt Deeter[2], Dylan B. A. Jones[3], Avelino F. Arellano[4], Daven K. Henze[5]

[1]Jet Propulsion Laboratory, California Institute of Technology, Pasadena, CA, USA
[2]National Center for Atmospheric Research, Boulder, CO, USA
[3]Department of Physics, University of Toronto, Toronto, ON, Canada
[4]Department of Hydrology and Atmospheric Sciences, University of Arizona, Tucson, AZ, USA
[5]Department of Mechanical Engineering, University of Colorado, Boulder, CO, USA



**Abstract**
Long-term measurements from satellites and surface stations have demonstrated a
decreasing trend of tropospheric carbon monoxide (CO) in the Northern Hemisphere over the past
decade. Likely explanations for this decrease include changes in anthropogenic, fires, and/or
biogenic emissions or changes in the primary chemical sink hydroxyl radical (OH). Using
remotely sensed CO measurements from the Measurement of Pollution in the Troposphere
(MOPITT) satellite instrument, in-situ methyl chloroform (MCF) measurements from World Data
Centre for Greenhouse Gases (WDCGG), and the adjoint of the GEOS-Chem model, we estimate
the change in global CO emissions from 2001-2015. We show that the loss rate of MCF varies by
0.2% in the past 15 years, indicating that changes in global OH distributions do not explain the
recent decrease in CO. Our two-step inversion approach for estimating CO emissions is intended
to mitigate the effect of bias errors in the MOPITT data as well as model errors in transport and
chemistry, which are the primary uncertainties when quantifying CO emissions using these
remotely sensed data. Our results confirm that the decreasing trend of tropospheric CO in the
Northern Hemisphere is due to decreasing CO emissions from anthropogenic and biomass burning
sources. In particular, we find decreasing CO emissions from the United States and China in the
past 15 years, unchanged anthropogenic CO emissions from Europe since 2008, and likely a
positive trend from India and southeast Asia, in contrast to recently reported results. We find
decreasing trends of biomass burning CO emissions from boreal North America, boreal Asia and
South America, but little change over Africa. The inconsistency between our analysis with recent
study suggests more efforts are needed for robust conclusion about the variation of anthropogenic
CO emissions for India and Southeast Asia.





## 52  1. Introduction

Tropospheric CO is a product of incomplete combustion and a byproduct of the oxidation
of hydrocarbons. It plays a key role in atmospheric chemistry because it is the main sink for OH,
and an important precursor for tropospheric ozone ($O_3$). Recent studies demonstrated significant
change in tropospheric CO abundance in the past decade. Using Atmospheric Infrared Sounder
(AIRS) CO measurements, Warner et al. (2013) indicated that Northern Hemispheric CO mixing
ratio decreased by 1.28 ppb/year in the period of 2003-2012. Worden et al. (2013) demonstrated
Northern Hemispheric CO column measurements from MOPITT show a decrease of ~0.92%/year
in the period of 2000-2011. Using observations from Mt. Bachelor Observatory, Gratz et al. (2015)
also show a negative trend of CO concentration by 1.9%/year in the period of 2004-2013.
However, the reason for the large variation of tropospheric CO abundance is still unclear; for
example, Strode et al. (2016) found decreases in modeled CO abundance over North America and
Europe, but increases over China, based on bottom-up emissions.
The budget of tropospheric CO is determined by its sources and sinks. There is currently
much effort focused on accurately quantifying emissions of CO. For fossil fuels and biofuels,
energy consumption statistics and emission factors are usually used to construct the emission
inventories (e.g. Streets et al. 2006; Ohara et al. 2007; Zhang et al. 2009; Zhao et al. 2012).
Biomass burning emissions are commonly calculated as the product of burned area, fuel loads,
combustion completeness and emission factors (e.g. van der Werf et al. 2006, 2010; van Leeuwen
and van der Werf 2011). Because of the large uncertainties in the emission inventories, space-
based remotely sensed measurements and surface/aircraft in-situ observations have been
assimilated to provide "top-down" constraints on CO emissions (e.g., Arellano et al., 2006;
Chevallier et al. 2009; Jones et al., 2009; Kopacz et al., 2010; Jiang et al., 2011; Fortems-Cheiney





et al. 2011; Hooghiemstra et al. 2012; Miyazaki et al. 2015). In a recent study, Yin et al. (2015)
constrained global CO emissions for the period 2002-2011 to investigate the possible reasons for
the decreasing CO abundance in the Northern Hemisphere. Using MOPITT column data (version
6J) over the whole globe, Yin et al. (2015) indicate that the negative trend in the Northern
Hemisphere is driven by decreasing anthropogenic emissions from North America, Europe and
China, similar to our result.

The major sink of tropospheric CO is OH. Because of its high variability and short lifetime

(about one second), it is difficult to assess the spatial and temporal variation of global OH through
direct measurements (Spivakovsky et al. 2000; Lelieveld et al. 2004). Alternatively, Montzka et
al. (2011) demonstrated small interannual variability of global OH for the period 1997-2007 by
using the loss rate of MCF as a proxy. The measurements of MCF are assimilated in recent CO
inversion studies to provide updated OH (e.g. Fortems-Cheiney et al. 2011, 2012; Yin et al. 2015),
but the estimates are adversely affected by the sparse distribution of measurements.

The objective of this work is to investigate the dominant reasons for the decreasing CO

trend in the Northern Hemisphere, and to provide updated CO emission estimates for model studies.
Our approach for estimating emissions is intended to reduce the effects of model errors of transport
and chemistry, as well as bias errors in the data, on our conclusions about CO emissions; these are
the primary uncertainties that affect CO emissions estimates. For example, bias errors as a function
of latitude in MOPITT data can have a substantial impact on emissions estimates (Deeter et al.,
2014). Model errors of transport and chemistry will have variable and substantial effects on CO
emissions in different parts of the globe due to seasonal and latitudinal variations in convection,
advection, and boundary layer height (Jiang et al., 2013, 2015a, 2015b).

In order to suppress the influences from these systematic measurement and model





transport systematic biases, we performed a two-step inversion by combining sequential Kalman
Filter (Jiang et al. 2013, 2015a, 2015b) with four-dimensional variational (4D-Var) assimilation
(Henze et al. 2007) in this work, using the GEOS-Chem model. Instead of optimizing the CO
concentrations and emissions simultaneously (e.g. Fortems-Cheiney et al. 2011, 2012; Yin et al.
2015), our first step, the sequential Kalman Filter, modifies the atmospheric CO concentration
directly to provide low bias initial (monthly) and boundary (hourly) conditions, whereas the second
step (4D-Var) constrains CO emissions assuming perfect initial and boundary conditions. We also
apply bias corrections to MOPITT and compare the surface CO concentrations obtained by
constraining the model with either MOPITT profile, total column, or lower troposphere data to test
which data type provides the most accurate comparison with independent surface in-situ
measurements.

This paper is organized as follows: in Section 2 we describe the MOPITT instruments and

the GEOS-Chem model used in this work. In Section 3 we outline the inverse method. We then
investigate the long-term variations of global tropospheric OH and CO emissions in Section 4, and
we discuss the changes in tropospheric CO, and the contributions from emissions and
meteorological conditions. Our conclusions follow in Section 5.
**2. Observations and Model**
**2.1. MOPITT**

The MOPITT instrument was launched on December 18, 1999 on the NASA/Terra

spacecraft. The satellite is in a sun-synchronous polar orbit of 705 km and crosses the equator at
10:30 local time. The instrument makes measurements in a 612 km cross-track scan with a
footprint of 22 km $x$ 22 km, and provides global coverage every three days. The MOPITT data
used here were obtained from the joint (J) retrieval (V6J) of CO from TIR (4.7μm) and NIR (2.3μm)





radiances using an optimal estimation approach (Worden et al., 2010; Deeter et al., 2011). The
retrieved volume mixing ratios (VMR) are reported as layer averages of 10 pressure levels (surface,
900, 800, 700, 600, 500, 400, 300, 200 and 100 hPa). The relationship between the retrieved CO
profile and the true atmospheric state can be described as:
$$\hat{z} = z_a + A(z - z_a) + G\epsilon \tag{1}$$

where $z_a$ is the MOPITT a priori CO profile, $z$ is the true atmospheric state, $G\varepsilon$ describes the
retrieval error, and $A = \partial\hat{z}/\partial z$ is the MOPITT averaging kernel matrix, which gives the sensitivity
of the retrieval to the actual CO in the atmosphere. The MOPITT V6 data have been evaluated by
Deeter et al. (2014) using aircraft measurements from HIAPER Pole-to-Pole Observations (HIPPO)
and the National Oceanic and Atmospheric Administration (NOAA). For the TIR/NIR multi-
spectral retrievals, they found negative bias drift (-1.27%/year) at lower troposphere (800 hPa),
and positive bias drift (1.64%/year) at upper troposphere (200 hPa). The bias drift in the total
column is negligible (0.003%/year).

Figure 1 shows the comparison between MOPITT CO retrievals and HIPPO aircraft

measurements. The aircraft measurements are smoothed with MOPITT averaging kernels. The
comparison demonstrates a negative bias of MOPITT CO retrievals in the tropics and a positive
bias at the middle latitudes in the lower troposphere. Opposite bias is observed in the upper
troposphere. Similar latitude dependent biases in remote sensing retrievals have been revealed for
methane ($CH_4$) observations from Scanning Imaging Absorption Spectrometer for Atmospheric
Chartography (SCIAMACHY, Bergamaschi et al. 2007, 2009; Meirink et al. 2008), Greenhouse
Gases Observing Satellite (GOSAT, Turner et al. 2015), and CO observation from MOPITT
(version 4, Hooghiemstra et al. 2012). Similar to previous studies, we reduce the adverse effect of
the latitude dependent bias by applying latitude dependent correction factors to MOPITT CO


retrievals, based on the black solid line in Figure 1, which represents a 4-order polynomial curve
fitting (in a least-squares sense) for all data points.
**2.2. GEOS-Chem**
The GEOS-Chem global chemical transport model (CTM) [www.geos-chem.org] is driven
by assimilated meteorological fields from the NASA Goddard Earth Observing System (GEOS-5)
at the Global Modeling and data Assimilation Office. For the simulations in this work, various
versions of GEOS meteorological fields are used, including GEOS-4 (2000-2003), GEOS-5 (2004-
2012) and GEOS-FP (2013-2015). We use version v35j of the GEOS-Chem adjoint, which is based
on v8-02-01 of the forward GEOS-Chem model, with relevant updates through v9-02-01. Our
analysis is conducted at a horizontal resolution of 4°x5° with 47 vertical levels and employs the
CO-only simulation in GEOS-Chem, which uses archived monthly OH fields from the full
chemistry simulation. The OH fields used in this work are from GEOS-Chem version v5-07-08,
with a global annual mean OH concentration of $0.99x10^6$ molec/cm$^3$ (Evans et al. 2005). The
potential long-term variation of global tropospheric OH is evaluated in section 4.
The global anthropogenic emission inventory is from EDGAR 3.2FT2000 (Olivier et al.,
2001), but are replaced by the following regional emission inventories: the US Environmental
Protection Agency National Emission Inventory (NEI) for 2008 in North America, the Criteria Air
Contaminants (CAC) inventory for Canada, the Big Bend Regional Aerosol and Visibility
Observational (BRAVO) Study Emissions Inventory for Mexico (Kuhns et al. 2003), the
Cooperative Program for Monitoring and Evaluation of the Long-range Transmission of Air
Pollutants in Europe (EMEP) inventory for Europe in 2000 (Vestreng et al. 2002) and the INTEX-
B Asia emissions inventory for 2006 (Zhang et al. 2009). Biomass burning emissions are based on
the Global Fire Emission Database (GFED3, van der Werf et al. 2010). The a priori biomass





burning emissions in Sep-Nov 2006 were applied to Sep-Nov 2015 over Indonesia. Additional CO
sources come from oxidation of methane and biogenic volatile organic compounds (VOCs) as
described in previous studies (Kopacz et al. 2010; Jiang et al. 2013). The biogenic emissions are
simulated using the Model of Emissions of Gases and Aerosols from Nature, version 2.0
(MEGANv2.0, Guenther et al. 2006). The distribution of the annual mean CO emissions for 2001-
2015 is shown in Figure 2. The annual global sources are 892 Tg CO from fossil fuel, biofuel and
biomass burning, 623 Tg CO from the oxidation of biogenic VOCs, and 876 Tg CO from the
oxidation of CH$_4$.
**3. Inversion Approach**

We use the 4D-var data assimilation system in GEOS-Chem (Henze et al. 2007) to

constrain the CO sources. In this approach, we minimize the cost function defined as:
$$J(x) = \sum_{i=1}^{N} (F_i(x) - z_i)^T S_\Sigma^{-1} (F_i(x) - z_i) + (x - x_a)^T S_a^{-1} (x - x_a)$$

where $x$ is the state vector of CO emissions, $N$ is the number of MOPITT observations that are
distributed in time over the assimilation period, $z_i$ is a given MOPITT measurement, and $F(x)$ is
the forward model. The error estimates are assumed to be Gaussian, and are given by $S_\Sigma$, the
observational error covariance matrix, and $S_a$, the a priori error covariance matrix, respectively.
The Gaussian assumption excludes important systematic errors, such as biases in OH distribution,
long-range transport and satellite retrievals in the cost function. Due to lack of meaningful
information about the systematic errors, we assume a uniform observation error of 20% without
spatial correlation. The combustion CO sources (fossil fuel, biofuel and biomass burning) and the
oxidation source from biogenic VOCs are combined together, assuming a 50% uniform a priori
error. We optimize the source of CO from the oxidation of CH$_4$ separately as an aggregated global





source, assuming an a priori uncertainty of 25%.
Because the 4D-var optimization scheme does not store the full Hessian matrix, we do not
construct the a posteriori error covariance matrix, which is the inverse of the Hessian. As opposed
to earlier studies using surface measurements, the high spatial density of measurements from
satellite instruments can effectively suppress the contribution from random errors in the cost
function, leaving systematic errors as the critical factor in the uncertainty. As shown by Heald et
al. (2004), different assumptions about the inversion configuration can produce differences in the
source estimates that are significantly larger than the a posteriori errors.
Removing the bias in initial conditions is essential for inverse analysis, and can be
performed with various data assimilation techniques. Model simulations driven by optimized
emissions can provide good initial conditions (e.g. Gonzi et al. 2011; Bruhwiler et al. 2014; Deng
et al. 2014; Houweling et al. 2014). Alternatively, tracer concentrations can be modified directly
to avoid the effect from long-range transport error (e.g. Kopacz et al. 2009; Jiang et al. 2013,
2015a). There are also efforts to optimize emissions and concentrations simultaneously (e.g.
Fortems-Cheiney et al. 2011, 2012; Bergamaschi et al. 2013; Yin et al. 2015), however, the
contributions from emissions and concentrations to model bias may be hard to be distinguished.
Figure 3 shows the methodology of our assimilation system. Following our previous studies (Jiang
et al. 2013, 2015a, 2015b), we produce initial conditions at the beginning of each monthly
assimilation window by assimilating MOPITT data using a sequential Kalman filter. For the results
presented here, the Kalman filter assimilation was carried out from March 1, 2000 to December

31, 2015.

Systematic errors have critical influences on inverse analysis. Jiang et al. (2013) found that
the modeled CO concentration from a 10-day forecast simulation have large discrepancy with



assimilated CO fields, because of bias in model convective transport. Jiang et al. (2015a)
demonstrated that free tropospheric CO is more susceptible to the influences of OH bias than lower
tropospheric CO due to the process of long-range transport. On the other hand, Jiang et al. (2015b)
indicated that regional inversions have more advantages than global inversions because the
boundary conditions can be better controlled. They demonstrated that the systematic bias
associated with North American CO emissions due to OH distribution can be reduced by up to 50%
with optimized boundary conditions. Similar optimization on the boundary condition can also be
employed in global model, for example, Pifster et al. (2005) constrained biomass burning CO
emissions from boreal North America with optimized CO fields outside the impacted region.

In order to reduce the effects of systematic errors, we designed a two-step inversion to

enhance the contributions from local emissions to the discrepancy between model and data, while
keeping the influence from long-range transport as low as possible due to sources of uncertainties
(e.g. emission uncertainty in the upstream continent, uncertainties in the chemical sink and
convective transport in the transport pathway), that are difficult to quantify. As shown in Figure 3,
we define the ocean scene (red grids) as boundary conditions. In the first step of our inverse
analysis, sequential Kalman filter assimilation, we directly modify CO concentrations without any
change to emissions in order to provide an optimized CO fields as consistent as possible with
MOPITT. In the second step, the optimized CO fields are used to rewrite CO concentrations over
the ocean every hour, while 4D-var inversion is employed to constrain CO emissions, without any
change on CO distribution over ocean. Only MOPITT data over land (white grids) were
assimilated to constrain CO emissions in the second step. With the fixed/optimized boundary
conditions, the global inversion system has been converted to a combination of several regional
inversions. Consequently, the emission and transport errors from one continent (e.g. North



America) will not affect the emission estimation of another continent (e.g. Europe).

## 4. Results and Discussion

### 4.1. Long-term variation of global tropospheric OH

The distribution of tropospheric OH has significant influence on the inverse analysis of CO
emissions (Jiang et al. 2011). Various approaches have been employed to improve the OH
distribution in previous studies. Jiang et al. (2013) assimilated MOPITT CO retrievals in full
chemistry model simulation to provide updated OH fields. Miyazaki et al. (2015) demonstrated
that assimilation of Tropospheric Emission Spectrometer (TES) $O_3$, Ozone Monitoring Instrument
(OMI) $NO_2$, and MOPITT CO can provide a better description of tropospheric OH. There are also
recent efforts that have assimilated surface in-situ MCF measurements (Fortems-Cheiney et al.
2011, 2012; Yin et al. 2015). However, because of the uncertainties in model chemistry schemes,
potential bias drifts in satellite remotely sensed observation, and sparse distribution of surface in-
situ measurements, OH abundances provided by these approaches may not be ideal for the
estimation of long-term CO variation.
Emissions of MCF are regulated by the Montreal Protocol agreement. The loss rate of MCF
has become a good tool to evaluate the variation of tropospheric OH (e.g. Bousquet et al. 2005;
Prinn et al. 2005; Montzka et al. 2011). Using the same approach as Montzka et al. (2011), we
assess the variation of tropospheric OH in the period of 2001-2015. Figure 4a shows the locations
of WDCGG sites with MCF measurements, and   Figure 4b shows the global mean MCF
concentration in the past 15 years. Similar as Montzka et al. (2011), our result shows a exponential
decrease of MCF concentration. The loss rate of MCF, derived from 12-month apart of monthly
means [e.g., $\ln(MCF_{Jan2007}/MCF_{Jan2006})$] varies by 0.2% in the past 15 years (Figure 4c). The
interannual variation is more likely due to the sparsity and discontinuity of measurements.




The small variation of loss rate of MCF demonstrates that the long-term variation of global
mean OH distributions is negligible in the past 15 years. Consequently, the decreasing trend of
tropospheric CO in North Hemisphere is driven by decreasing CO sources, rather than sinks. For
this reason, the default monthly OH fields of GEOS-Chem model (Evans et al. 2005), without
interannual variability, are used in this work to constrain the long-term variation of CO emissions.
Because the abundances of tropospheric OH have large regional discrepancies (e.g. Jiang et al.
2015a), it is possible that the actual OH is more variable at regions lacking MCF measurements
(e.g. India and southeast Asia). Futhermore, the magnitude and seasonality of the default monthly
OH fields could also have uncertainty. Consequently, the magnitude of CO emissions in our
analysis may still be affected by biases in OH, although the two-step assimilation system is
designed to suppress their influence.
**4.2. Long-term variation of global CO emissions**
In this work, we performed monthly inversions for the period of 2001-2015, using
MOPITT column, profile and lower tropospheric profile (lowest three retrieval levels) data to
investigate the influences associated with vertical sensitivity of satellite instrument and model
transport error. Figure 5 shows the CO emission trends for 2001-2015 constrained by these
different datasets. For anthropogenic sources, all three analysis show significant emission
reduction from North America, Europe and China. The emission estimates constrained with
MOPITT column and profile data demonstrate increasing CO emissions from India and Southeast
Asia. Conversely, the emission estimate constrained with MOPITT lower tropospheric profile data
shows a decreasing trend in this region, and this decreasing trend is also obtained by Yin et al.
(2015). As shown in Jiang et al., (2013), errors in model convection in this region have a large
effect on CO emissions estimates, and information about the vertical profile of CO has a stronger



influence on the results.

For biomass burning sources, we found a negative trend over boreal North America, boreal

Asia and South America, and a positive trend over Indonesia that is primarily due to the strong
impacts of El Nino in 2006 and 2015 on biomass burning in this region (e.g. Field et al., 2016).
Our results for biogenic VOCs are inconclusive; the emission estimates constrained with MOPITT
column and profile data show moderate positive trends in the tropics, and slight negative trends in
mid-latitude regions, whereas the emission estimate constrained with MOPITT lower tropospheric
profile data shows a negative trend globally.

Figure 6a shows the regional variation of anthropogenic emissions from the United States

(US). The emission estimates constrained with MOPITT column and profile data match very well
with the a priori emissions, whereas the emission estimate constrained with MOPITT lower
tropospheric profile data is much higher. All three analyses demonstrate a significant emission
reduction over our study period. As shown in Table 1, the total anthropogenic CO emission
(constrained with MOPITT profile data) from US is 56.8 Tg in 2015, which is 35% lower than that
in 2001 (87.7 Tg). Figure 7a shows the monthly mean CO concentrations from WDCGG stations
in US, which demonstrates a similar decreasing trend as our analysis. The decreasing trend is
consistent with the US Environmental Protection Agency (EPA) Emissions Trends Data
(https://www.epa.gov/air-emissions-inventories/air-pollutant-emissions-trends-data),    and    other
observation records for western US (Gratz et al. 2015), southeast US (Hidy et al. 2014) and North
Atlantic (Kumar et al. 2013).

Figure 6b shows the regional variation of anthropogenic emissions from Europe. All three

analyses show an underestimation of a priori emissions, suggesting the CO emissions in the EMEP
inventory are too low. Our results show that anthropogenic emissions decrease during the period



of 2001-2007, but are almost unchanged in the following years, which is consistent with the
observations from WDCGG stations (Figure 7b). Recent studies (Hilboll et al. 2013; Schneider et
al. 2015) showed that $NO_2$ over Europe from SCIAMACHY is decreasing in the period of 2002-
2008, and almost unchanged in the period of 2008-2011. Henschel et al. (2015) indicated that the
unchanged $NO_2$ over Europe could be caused by European emissions that are failing to achieve
the expected reduction standards. Because anthropogenic CO and $NO_2$ share some of the same
combustion sources, it is possible that the unchanged CO emission in our analysis is also due to a
failure of emission controls.

Figure 6c shows the regional variation of anthropogenic emissions from east China. We

found Chinese anthropogenic emissions are increasing in the period of 2001-2004. Accompanied
with the global economy recession, the total anthropogenic CO emission (constrained with
MOPITT profile data) from east China decreases to 175.4 Tg in 2008, which is 15% lower than
that in 2004 (205.6 Tg). Our analysis shows a temporary increase of Chinese emissions in 2009
(185.9 Tg), followed by continuous decrease. The total Chinese anthropogenic CO emission is
159.0 Tg in 2015, which is 7% lower than that in 2001 (170.4 Tg). Using surface in-situ
measurements at Hateruma Island, Tohjima et al. (2014) constrained CO emissions from China
for the period 1999-2010. They found Chinese CO emission increases from 1999-2004, and
decreases since 2005. Using a "bottom-up" approach, recent studies (Zhao et al. 2012; Xia et al.
2016) indicated that the growth trend of Chinese CO emissions has been changed since 2005
because of improvements in energy efficiency and emission control regulations (e.g. Liu et al.
2015). Figure 7c shows the observation records from 2 stations in the East China outflow region,
which demonstrate similar variations.

Figures 6d-6e show the regional variation of anthropogenic emissions from India and



Southeast Asia. The emission estimates constrained with MOPITT column and profile data
demonstrate significant positive trend in our study period, whereas the emission estimate
constrained with MOPITT lower tropospheric profile data shows a decreasing trend. Schneider et
al. (2015) showed that $NO_2$ over south Asia from SCIAMACHY is increasing in the period of
2003-2011. Using OMI $NO_2$ measurements, recent studies (e.g., Duncan et al. 2016) demonstrated
that $NO_2$ over India has a positive trend during 2005-2015. Observations from Cape Rama (CRI)
station (Figure 7d) demonstrate that CO concentration in 2010-2013 is significantly higher than
that in 2001-2002. For these reasons, we have more confidence in our results that indicate
increasing anthropogenic CO emissions from India and Southeast Asia in the past 15 years. The
trend based on the MOPITT lower-tropospheric data is incorrect because of model error in
convection and boundary layer height in this dynamically varying region, and the negative bias
drift in MOPITT lower tropospheric retrievals (Deeter et al., 2014). The total anthropogenic CO
emission (constrained with MOPITT profile data) from India and Southeast Asia is 130.4 Tg in
2015, which is 34% higher than that in 2001 (97.5 Tg). It should be noted that the inconsistency
between our analysis with Yin et al. (2015) suggests more studies are needed for robust conclusion
about the variation of anthropogenic CO emissions for this region.

Although our inverse analysis (constrained with MOPITT profile data) suggests similar

anthropogenic CO emissions from East China in 2008 and 2014, Figure 7c demonstrates that mean
CO concentrations over the outflow region of East China are 6 ppb higher in 2014 compared to
2008. Our previous study (Jiang et al. 2015c) indicated that anthropogenic emissions from India
and southeast Asia have an important influence on pollutant concentrations in the east China
outflow region. It is possible that the increase of CO concentration observed by WDCGG stations
in this region is caused by the significant increase of anthropogenic CO emission from India and



southeast Asia. In the most recent 5 years (2011-2015), our results (constrained with MOPITT
profile data) suggested a 20.5 Tg emission reduction from East China, and a 10.1 Tg emission
increase from India and Southeast Asia. Assuming a fixed emission growth rate, projected
anthropogenic CO emissions from India and Southeast Asia will overtake Chinese emissions in
2020, resulting in  serious socioeconomic issues on both local and global scales.

Figure 8 shows the regional variation of biomass burning emissions. There are significant

decreasing trends in three regions (i.e. boreal North America, boreal Asia, and South America).
Our results show high biomass burning emissions from boreal North America (mainly Alaska and
western Canada) in 2004 (Figure 8a), which have been reported by previous studies (e.g. Pfister et
al. 2005; Turquety et al. 2007), and also from  boreal Asia during 2001-2003 (Figure 8b) due to
significant fire activity in Siberia (e.g., Yurganov et al., 2005,  Stroppiana et al., 2010). For South
America (Figure 8c), we found higher biomass burning emissions in the periods of 2004-2007 and
2010, consistent with fire activity reported in previous studies (e.g. Hooghiemstra et al. 2012;
Bloom et al. 2015).

Figure 8d shows the regional variation of biomass burning emissions from Africa. The fire

activities in Africa demonstrates obvious seasonality: peak in boreal winter for Northern
Hemispheric Africa, and in austral winter for Southern Hemispheric Africa. Similar to previous
studies (e.g. Chevallier et al. 2009; Tosca et al. 2015), there is no obvious emission trend in Africa
in the past 15 years. This is also consistent with the burned area trends described by Andela et al.
(2014) which show opposite  directions for Northern Africa (decreasing) versus Southern Africa
(increasing) and would have cancelling effects in the trend for the continent as a whole.

Our results exhibit two strong biomass burning events in Indonesia, 2006 and 2015,

individually (Figure 8e). Previous studies (e.g. Logan et al. 2008; Zhang et al. 2011; Worden et al.



2013b, 2013c, Field et al., 2016) demonstrate the direct relationship between strong Indonesian
fires and El Niño. More recent studies (Huang et al. 2014; Inness et al. 2015) confirm low biomass
burning activities in Indonesia in the period of 2007-2012. CO emissions from the Indonesian fires
associated with the 2015 El Niño were 92 Tg, (for October, 2015, as constrained with MOPITT
profile data), and were about three times higher than the October 2006 El Nino driven fire
emissions (32 Tg). Not including the 2015 El Niño driven fires,  our analysis indicates a negative
trend of global biomass burning emissions in the past 15 years, as shown in Figure11f.
**4.3. Changes in tropospheric CO during 2001-2015**

In this section, we evaluate our inversion results using independent long-term surface in-

situ measurements from WDCGG stations. Figure 9a shows the annual trend of surface CO
concentration for 2001 – 2015 from WDCGG sites, and from model simulations driven with a
priori emissions. Most WDCGG sites exhibit negative trends in the past 15 years, confirming the
decreasing trend of global tropospheric CO, which is consistent with satellite observations (e.g.
Warner et al. 2013; Worden et al. 2013). There are also stations with positive trends, for example,
Tae-ahn Peninsula (TAP, Korea), Ascension Island (ASC, equtorial Atlantic Ocean), Cape Rama
(CRI, India),  Bukit Koto Tabang (BKT, Indonesia) and Cape Grim (CGO, Australia). Globally,
the a priori model simulation is in reasonable agreement with WDCGG measurements: both show
negative trends in middle/high latitude, and positive trends in some tropical regions. However,
there are noticable discrepancies, for example, the surface observation from Yonagunijima (YON,
east China sea) shows a negative trend in our study period, suggesting decreasing trend from
Chinese CO emission, whereas the a priori simulation demonstrates significant positive trend.

Figure 9b-9d show the model simulations driven with a posteriori emissions. The a

posteriori emissions constrained with MOPITT lower tropospheric profile data (Figure 9d) results



in unrealistic large CO reduction, which could be caused by the negative bias drift of MOPITT
retrievals at lower troposphere (Deeter et al. 2014) and the influence from possible variability in
model convective transport. The a posteriori emissions constrained with MOPITT column and
profile data have similar comparisons. For example, both of them suggest a negative trend over
east China, consistent with observations from YON, and positive trend over northeast Asia,
consistent with observations from TAP.

In order to better compare the discrepancy between model simulation and surface

observations, Figure 9e-9g show the improvement due to a posteriori emissions, derived by
$abs(Trend_{aposteriori} - Trend_{WDCGG})$ - $abs(Trend_{apriori} - Trend_{WDCGG})$. Blue (red) means the a posteriori
emissions improves (degrades) the agreement with WDCGG measurements compared to the
simulated surface CO using a priori emissions, while white indicates no change from the prior. As
shown in Figure 9f, the CO emissions constrained with MOPITT profile data improved the model
simulation for most WDCGG sites in the Northern Hemisphere. The a posteriori emissions
constrained with MOPITT column data are somewhat worse, particularly over Europe, while CO
emissions constrained with MOPITT profile data over Europe give improved comparisons to
WDCGG surface CO measurements. Worden et al. (2010) demonstrated that the degrees of
freedom for signal (DFS) of MOPITT multi-spectral profile retrievals (TIR+NIR) is about 1.5-2.0
over land, which is reduced to about 1 DFS when converted to a total column.  This reduction in
vertical information in MOPITT column data can affect the the reliability of inverse analysis
results (Jiang et al., 2015a). It should be noticed that the vertical correlation in model simulation
is not considered in our assimilation, which could be another possible reason for this discrepancy.

Figure 10a-10d show the long-term mean value of surface CO concentration for 2001 –

2015 from WDCGG sites, and model simulations driven with a priori and a posteriori emissions.


419 All simulations provide similar results for long-term mean value. Figure 10e-10g show the

420 improvement due to a posteriori emissions, derived by abs($CO_{aposteriori} - CO_{WDCGG}$) - abs($CO_{apriori}$

421 - $CO_{WDCGG}$). Figure 10f demonstrates that CO emissions constrained with MOPITT profile data

422 improved the model simulation in about half of the sites in the Northern Hemisphere, whereas the

423 a posteriori emissions constrained with MOPITT column data are somewhat worse (Figure 10e).

424 Evaluating modeled tracer concentrations using surface in-situ measurements is more challenging

425 than evaluating long-term trends. In a recent study, Schnell et al. (2015) evaluate surface $O_3$

426 concentrations simulated by multi-models for North America and Europe. They found most

427 models can provide good simulations for the patterns of $O_3$ but cannot reproduce the magnitude.

428 Important sources of uncertainty include the representation error (e.g. Chang et al. 2015; Kharol

429 et al. 2015) and vertical mixing of boundary layer (e.g. Castellanos et al. 2011; Cuchiara et al.

430 2014).

431  Because our a posteriori simulation, particularly using emissions constrained with

432 MOPITT profile data, results in significant improvement in the long-term trend, and moderate

433 improvement in the mean value, we believe these a posteriori estimates provide a better description

434 for the long-term variation of global CO emissions. A remaining question is to explore how

435 changes in meterological conditions affect the long-term variation. By fixing CO emissions to

436 2001 levels, Figure 11a-11b show the long-term trend of modeled surface and column CO during

437 2001-2015, due only to changes in meterological conditions. At the surface level (Figure 11a), we

438 found changes in meterology result in a moderate positive trend in the Northern Hemisphere,

439 particularly, over northeast Asia, consistent with observation records from the TAP station; and

440 significant positive trend in tropics, consistent with observation record from ASC station. On the

441 other hand, the influence of meterological conditions on column CO (Figure 11b) is much weaker.



The discrepancy between surface and column CO suggests the possible contribution from variable
convective transport, which could be associated with changes in the frequency of deep convection
(Tan et al. 2015) or the change from El Niño to La Niña in our study period (Andela et al. 2014).
In order to assess the influence of various versions of the meterological fields (i.e. GEOS-4, GEOS-
5 and GEOS-FP) on the trend analysis, we reploted (not shown) Figure 11a-11b for the period
2004-2012 with GEOS-5 meterological fields, and obtained similar significant positive trend in
tropics, which suggests limited influence from meterological field version differences on the trend
analysis.

Figure 11c-11h show the variation of global tropospheric CO due to changes in emissions.

Yin et al. (2015) indicated that the negative trend of tropospheric CO in the Northern Hemisphere
is driven by decreasing anthropogenic emissions from North America, Europe and China. Along
with reductions in anthropogenic emissions (Figure 11c, 11d), we found the decrease of biomass
burning emissions from boreal North America and boreal Asia (Figure 11e, 11f) to be an important
factor for this negative trend. In constrast to the emission reduction from North America, Europe
and China, we found increasing anthropogenic emissions from India and southeast Asia, which
result in a pronounced positive trend of tropospheric CO, while Yin et al. (2015) obtain a negative
trend for this region. This discrepancy requires further study and we will need to test the relative
importance of the primary differences in our methods, i.e., models and inversion approaches,
climatological OH (this study) vs. assimilated surface measurements of $CH_4$ and MCF to update
OH (Yin et al.) and the use of MOPITT profile vs. column CO retrievals (Yin et al., assimilate
only column CO).
**5. Summary**

The objective of this work is to investigate the dominant reasons for the observed variation



of global tropospheric CO over the past 15 years, and to provide updated CO emission estimates
for model studies. In particular, we use a combination of MOPITT CO measurements and surface
measurements of MCF to evaluate changes in the sources and sinks of atmospheric CO, with the
goal of explaining the observed decrease in CO concentrations. Our two-step approach for
estimating global CO emissions mitigates the effects of model errors from transport and chemistry,
as well as measurement bias error.

Using the same approach as Montzka et al. (2011), we assess the variation of tropospheric

OH (the primary CO sink) in the period of 2001-2015 using MCF measurements from WDCGG
stations. Our result demonstrates negligible variation of global tropospheric OH in the past 15
years, and consequently we suggest that the global sink of CO due to chemical loss through OH
has not likely changed during this time period. We therefore expect the decreasing trend of
tropospheric CO in North hemisphere (e.g. Warner et al. 2013; Worden et al. 2013; Gratz et al.
2015) to be driven by decreasing CO sources. Total anthropogenic CO emissions from the US
were 56.8 Tg in 2015, which are 35% lower than emissions in 2001 (87.7 Tg). Total anthropogenic
CO emissions from East China were 159.0 Tg in 2015, which are 7% lower than 2001 emissions
(170.4 Tg) and 23% lower than 2004 emissions (205.6 Tg). This pronounced decrease of emissions
from US and China is an indication of progress for fuel efficiency and emission control regulations.
Conversely, our results demonstrate that anthropogenic emissions from Europe decreased from
2001 to 2007 but are almost unchanged during 2008-2015. We also found a significant increase of
anthropogenic emissions for India and Southeast Asia. The total anthropogenic CO emission from
India and southeast Asia is 130.4 Tg in 2015, which is 34% higher than that in 2001 (97.5 Tg).
Assuming the same emission growth rate as 2011-2015, we expect that anthropogenic CO
emissions from India and Southeast Asia will be larger than Chinese emissions by 2020.



In a recent study, Yin et al. (2015) indicated that the decreasing tropospheric CO in the
Northern Hemisphere is caused by the decrease of anthropogenic emissions from North America,
Europe and China. We find that a decrease of biomass burning emissions from boreal North
America and boreal Asia is also an important contributor for the negative trend. Globally, our
analysis indicates a negative trend of biomass burning emissions in the past 15 years, except in
Indonesia due to the strong biomass burning event in 2015 associated with El Niño. Our results
demonstrate a significant decrease of biomass burning emissions from South America, which
could be associated with the reduction of deforestation in Brazil (Reddington et al. 2015), and the
predominant change from El Nino to La Nina in our study period (Andela et al. 2014). For Africa,
there is no obvious CO emission trend in the past 15 years, consistent with previous results
(Chevallier et al. 2009; Tosca et al. 2015; Andela et al., 2014). Our results are inconclusive in
characterizing the CO sources from oxidation of biogenic VOCs. More efforts are needed in the
future to better understand the mechanism for tropical CO emissions.
Our analysis highlights the importance of space-based instruments for monitoring changes
in global pollutant emissions. Our results demonstrate successful emission controls in US and
China over the past 15 years, and suggest that emission controls in Europe may need re-evaluation.
We also recommend more efforts in the future to better understand the regional and global effects
of increasing pollutant emissions from India and Southeast Asia.

**Acknowledgments**.
We thank the World Data Centre for Greenhouse Gases (WDCGG) for providing their CO
and MCF data. The National Center for Atmospheric Research (NCAR) is sponsored by the
National Science Foundation. The NCAR MOPITT project is supported by the National



Aeronautics and Space Administration (NASA) Earth Observing System (EOS) Program. The
MOPITT team also acknowledges support from the Canadian Space Agency (CSA), the Natural
Sciences and Engineering Research Council (NSERC) and Environment Canada, along with the
contributions of COMDEV (the prime contractor) and ABB BOMEM. MOPITT data sets used in
this    study    are    publicly    available    at    http://reverb.echo.nasa.gov    and    at
https://eosweb.larc.nasa.gov/datapool.

**Data availability**
The MOPITT data is available at ftp://l5eil01.larc.nasa.gov/MOPITT/MOP02J.006. The MCF and
CO measurements from WDCGG is available at http://ds.data.jma.go.jp/gmd/wdcgg/.

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

**Tables and Figures**



**Table 1**. Annual total anthropogenic CO emission in different regions, from 2001 to 2015,
constrained with MOPITT column, profile and lower tropospheric data. The region definition is
shown in Figure 2e.
**Figure 1**. Difference between MOPITT CO retrievals and HIPPO aircraft measurements. The
aircraft measurements are smoothed with MOPITT averaging kernels. The black solid line shows
the 4-order polynomial curve fitting, which is used to correct MOPITT data in this work.
**Figure 2**. (a-d) Mean a priori CO emissions from combustion sources and the oxidation of biogenic
VOCs and $CH_4$ from 2001 to 2015. The unit is $10^{12}$ molec/cm$^2$/sec. (e-f) Region definitions for (e)
anthropogenic and (f) biomass burning sources.
**Figure 3**. Schematic diagram for methodology of the assimilation system. Sequential Kalman
Filter was run from March 1 2000 to December 31 2015 to produce the optimized initial conditions
(monthly) and boundary conditions (hourly). Monthly 4-DVAR inversions were performed with
the optimized initial conditions. Only MOPITT data over land (white grids) were assimilated in
the 4-DVAR inversions, while the CO abundances over ocean (red grids) were defined as
boundaries and rewritten using the optimized hourly CO fields from Kalman Filter.
**Figure 4**. (a) Locations of WDCGG sites with MCF measurements. (b) Global mean MCF
concentration. (c) Exponential loss rate of MCF, derived from 12-month apart of monthly means
[e.g., ln($MCF_{Jan2007}$/$MCF_{Jan2006}$)]. The black solid line shows the 12-month mean value.
**Figure 5.** CO emission trends for 2001 – 2015, constrained with MOPITT column, profile and
lower tropospheric profile data. The months dominated by biomass burning emissions are excluded
from the trend calculation for anthropogenic and biogenic VOC emissions.
**Figure 6.** 12-month mean value of anthropogenic CO emissions (with unit Tg/month) for 2001 –
2015: a priori emission (green) and a posteriori emissions constrained with MOPITT column data
(black), MOPITT profile data (blue) and MOPITT lower tropospheric profile data (red). The green
dash line shows the monthly a priori anthropogenic CO emissions. The region definition is shown
in Figure 2e.
**Figure 7.** Monthly mean CO concentrations (green) and 12-month mean value (black) from
WDCGG stations for 2001 – 2015. (a) 15-station average in United States (b) 20-station average
in Europe (c) 2-station (YON and JMA) average in east China outflow (4) Cape Rama (CRI) in
India.
**Figure 8.** Monthly biomass burning CO emissions (with unit Tg/month) for 2001 – 2015: a priori
emission (green) and a posteriori emissions constrained with MOPITT column data (black),
MOPITT profile data (blue) and MOPITT lower tropospheric profile data (red). The region
definition is shown in Figure 2f.
**Figure 9.** Panels (a-d): long-term trend (annual) of surface CO concentration for 2001 – 2015 from
WDCGG sites, and model simulations driven with a priori and a posteriori emissions. Panels (e-
g): effect of a posteriori emissions, derived by abs(Trend$_{aposteriori}$ – Trend$_{WDCGG}$) - abs(Trend$_{apriori}$ -





Trend$_{WDCGG}$); blue (red) means the a posteriori emissions improves (degrades) the agreement with
WDCGG measurements compared to the a priori emissions, while white indicates no change from
the priori. Only stations with more than 10 year observations (the time range between the first and
last observations) during 2001-2015 are included.
**Figure 10.** Panels (a-d): long-term mean value of surface CO concentration for 2001 – 2015 from
WDCGG sites, and model simulations driven with a priori and a posteriori emissions. Panels (e-
g): effect of a posteriori emissions, derived by abs($CO_{aposteriori} - CO_{WDCGG}$) - abs($CO_{apriori}$ -
$CO_{WDCGG}$); blue (red) means the a posteriori emissions improves (degrades) the agreement with
WDCGG measurements compared to the a priori emissions, while white indicates no change from
the priori. Only stations with more than 10 year observations (the time range between the first and
last observations) during 2001-2015 are included.
**Figure 11.** Long-term trend (annual) of modeled surface and column CO for 2001 – 2015 with (a-
b) all emission sources are fixed at 2001 level. (c-d) variable anthropogenic emissions; (e-f)
variable biomass burning emissions; (g-h) variable biogenic VOCs emissions; The variable
emissions are constrained with MOPITT profile data.



| Years | MOPITT Column (Tg/year) | | | | MOPITT Profile (Tg/year) | | | | MOPITT Lower Profile (Tg/year) | | | |
|---|---|---|---|---|---|---|---|---|---|---|---|---|
| | United States | Europe | E China | India/SE Asia | United States | Europe | E China | India/SE Asia | United States | Europe | E China | India/SE Asia |
| 2001 | 87.8 | 71.6 | 165.7 | 102.2 | 87.7 | 77.3 | 170.4 | 97.5 | 112.9 | 92.0 | 215.7 | 136.1 |
| 2002 | 84.1 | 65.9 | 171.3 | 93.3 | 82.3 | 77.1 | 176.1 | 81.1 | 110.1 | 89.8 | 221.9 | 119.8 |
| 2003 | 80.8 | 65.3 | 178.8 | 95.4 | 80.4 | 74.5 | 189.2 | 88.5 | 103.6 | 87.0 | 218.1 | 121.9 |
| 2004 | 77.4 | 65.5 | 178.5 | 105.0 | 91.1 | 83.8 | 205.6 | 113.8 | 103.0 | 89.5 | 222.8 | 124.6 |
| 2005 | 72.7 | 64.6 | 178.6 | 104.3 | 82.6 | 79.4 | 200.6 | 116.8 | 92.7 | 84.5 | 215.3 | 126.2 |
| 2006 | 74.6 | 61.5 | 172.7 | 98.1 | 85.6 | 74.5 | 197.7 | 111.0 | 93.9 | 78.9 | 205.1 | 118.1 |
| 2007 | 73.7 | 56.5 | 177.1 | 105.8 | 84.0 | 67.9 | 200.9 | 113.2 | 90.9 | 71.8 | 208.1 | 119.4 |
| 2008 | 67.1 | 55.5 | 150.2 | 102.1 | 77.2 | 65.4 | 175.4 | 110.2 | 83.9 | 69.6 | 175.0 | 111.1 |
| 2009 | 66.0 | 54.8 | 162.0 | 105.7 | 74.5 | 65.1 | 185.9 | 118.3 | 78.0 | 67.0 | 184.5 | 115.1 |
| 2010 | 59.2 | 54.5 | 159.3 | 100.5 | 67.8 | 65.3 | 183.1 | 112.8 | 73.5 | 69.0 | 185.5 | 106.7 |
| 2011 | 53.5 | 52.9 | 153.2 | 107.4 | 60.5 | 63.1 | 179.5 | 120.3 | 63.0 | 65.6 | 175.7 | 107.5 |
| 2012 | 54.9 | 58.3 | 167.0 | 113.8 | 58.2 | 65.2 | 184.2 | 128.8 | 62.5 | 68.9 | 187.0 | 115.7 |
| 2013 | 54.3 | 62.6 | 160.4 | 120.9 | 56.7 | 68.8 | 171.2 | 131.3 | 61.8 | 73.8 | 176.8 | 114.6 |
| 2014 | 55.0 | 60.1 | 157.1 | 121.3 | 56.8 | 63.9 | 175.6 | 133.4 | 60.9 | 68.5 | 174.4 | 115.5 |
| 2015 | 55.1 | 61.4 | 145.1 | 115.6 | 56.8 | 66.9 | 159.0 | 130.4 | 59.5 | 69.3 | 160.5 | 109.2 |

**Table 1**. Annual total anthropogenic CO emission in different regions, from 2001 to 2015,
constrained with MOPITT column, profile and lower tropospheric data. The region definition
is shown in Figure 2e.






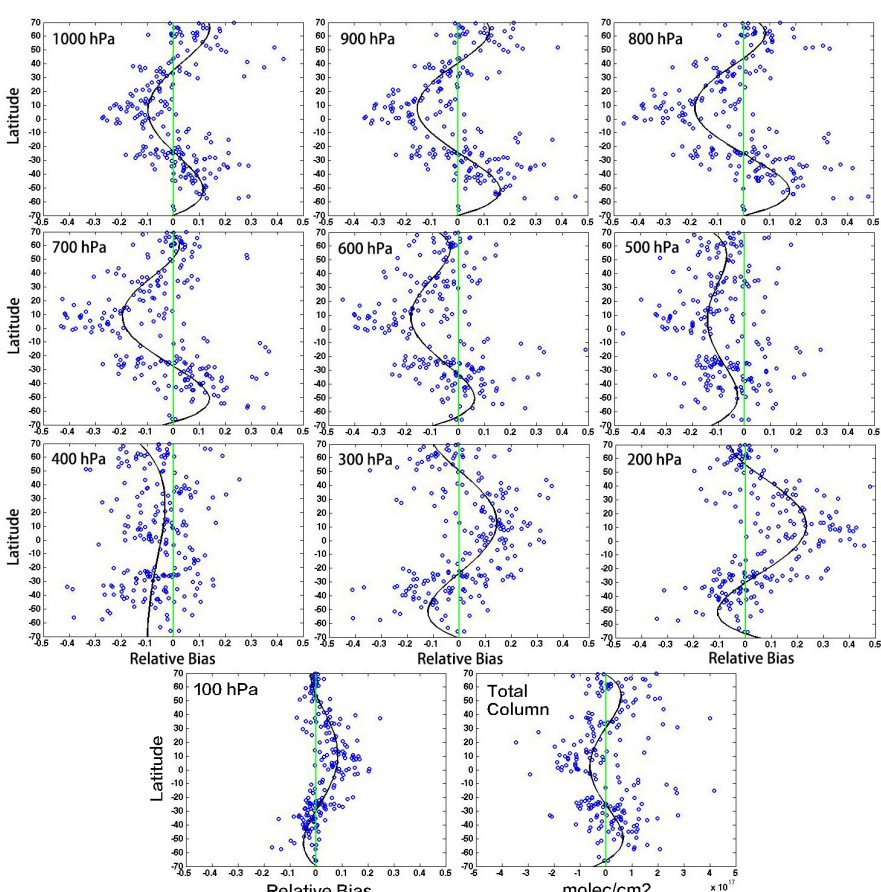


**Figure 1**. Difference between MOPITT CO retrievals and HIPPO aircraft measurements. The aircraft measurements are smoothed with MOPITT averaging kernels. The black solid line shows the 4-order polynomial curve fitting, which is used to correct MOPITT data in this work.




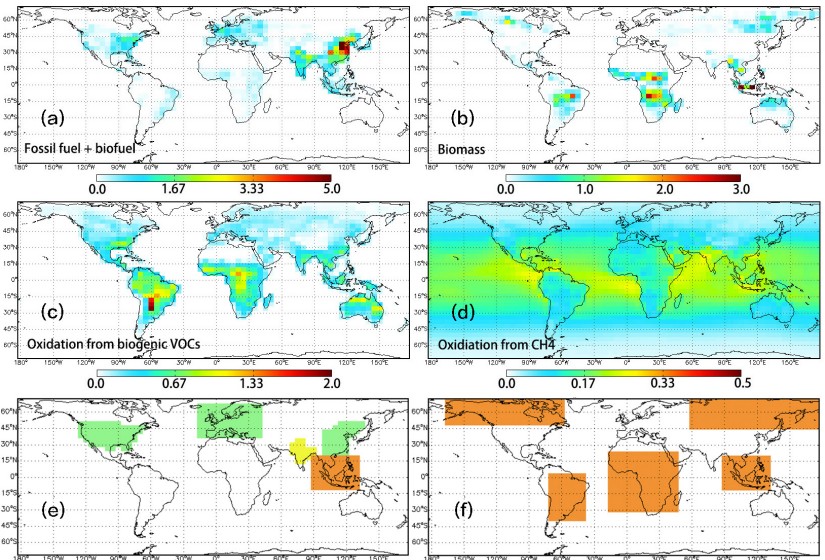

**Figure 2**. (a-d) Mean a priori CO emissions from combustion sources and the oxidation of
biogenic VOCs and $CH_4$ from 2001 to 2015. The unit is $10^{12}$ molec/cm²/sec. (e-f) Region
definitions for (e) anthropogenic and (f) biomass burning sources.

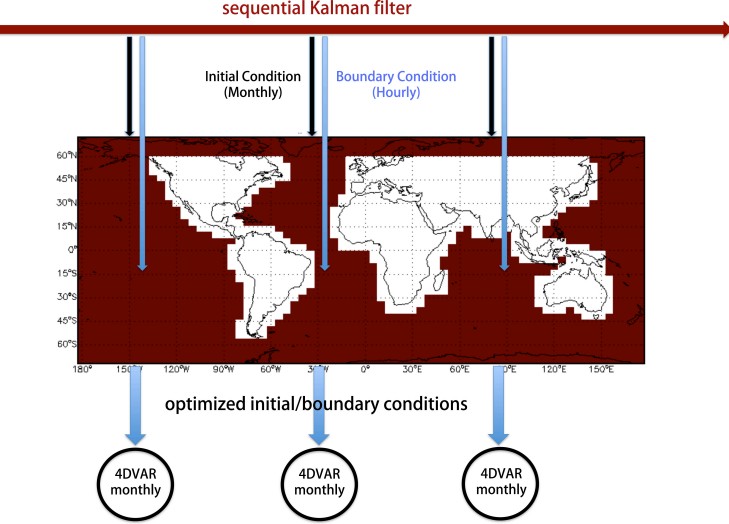


**Figure 3**. Schematic diagram for methodology of the assimilation system. Sequential Kalman
Filter was run from March 1 2000 to December 31 2015 to produce the optimized initial
conditions (monthly) and boundary conditions (hourly). Monthly 4-DVAR inversions were
performed with the optimized initial conditions. Only MOPITT data over land (white grids)
were assimilated in the 4-DVAR inversions, while the CO abundances over ocean (red grids)
were defined as boundaries and rewritten using the optimized hourly CO fields from Kalman
Filter.





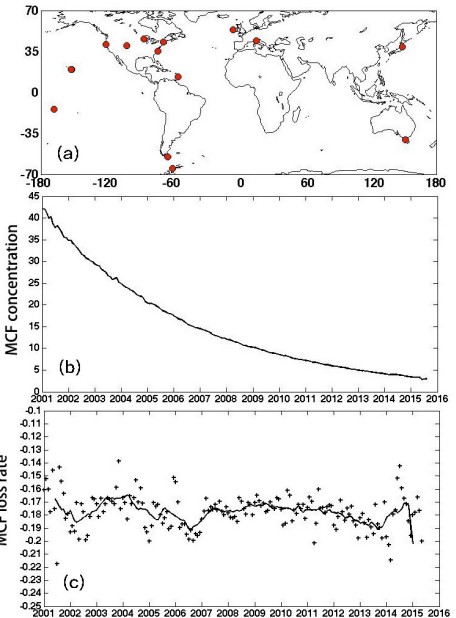


**Figure 4**. (a) Locations of WDCGG sites with MCF measurements. (b) Global mean MCF concentration. (c) Exponential loss rate of MCF, derived from 12-month apart of monthly means [e.g., $\ln(\text{MCF}_{\text{Jan2007}}/\text{MCF}_{\text{Jan2006}})$]. The black solid line shows the 12-month mean value.


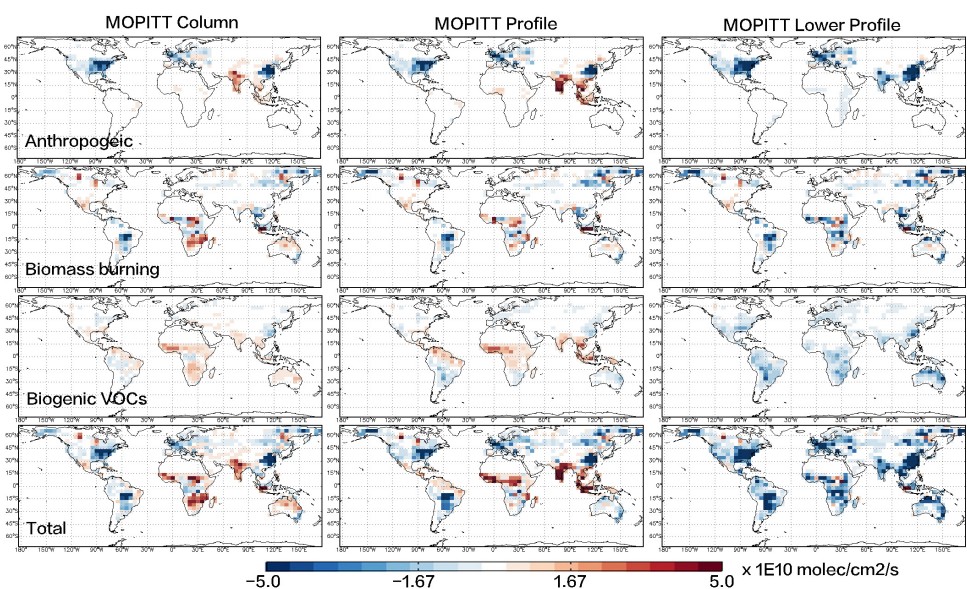


**Figure 5.** CO emission trends for 2001 – 2015, constrained with MOPITT column, profile and lower tropospheric profile data. The months dominated by biomass burning emissions are excluded from the trend calculation for anthropogenic and biogenic VOC emissions.




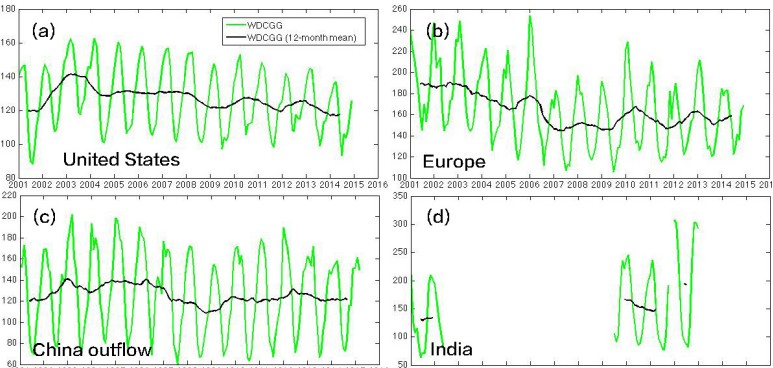


**Figure 6.** 12-month mean value of anthropogenic CO emissions (with unit Tg/month) for 2001
– 2015: a priori emission (green) and a posteriori emissions constrained with MOPITT column
data (black), MOPITT profile data (blue) and MOPITT lower tropospheric profile data (red).
The green dash line shows the monthly a priori anthropogenic CO emissions. The region
definition is shown in Figure 2e.

947

948

949

**Figure 7.** Monthly mean CO concentrations (green) and 12-month mean value (black) from
WDCGG stations for 2001 – 2015. (a) 15-station average in United States (b) 20-station
average in Europe (c) 2-station (YON and JMA) average in east China outflow (4) Cape Rama
(CRI) in India.






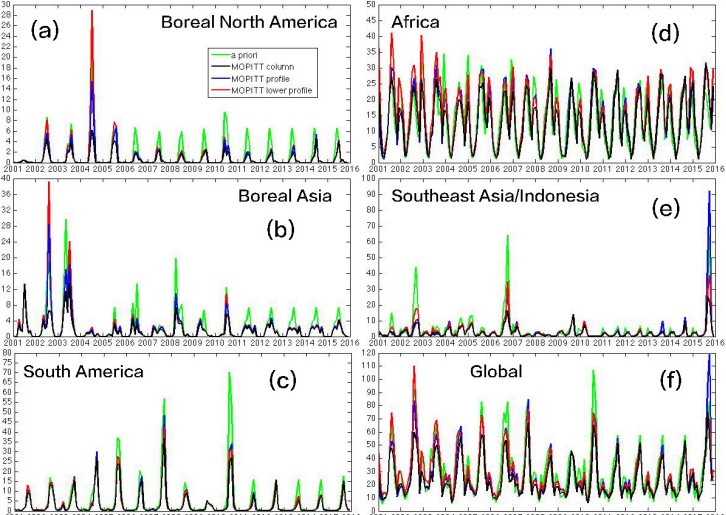


**Figure 8.** Monthly biomass burning CO emissions (with unit Tg/month) for 2001 – 2015: a priori emission (green) and a posteriori emissions constrained with MOPITT column data (black), MOPITT profile data (blue) and MOPITT lower tropospheric profile data (red). The region definition is shown in Figure 2f.





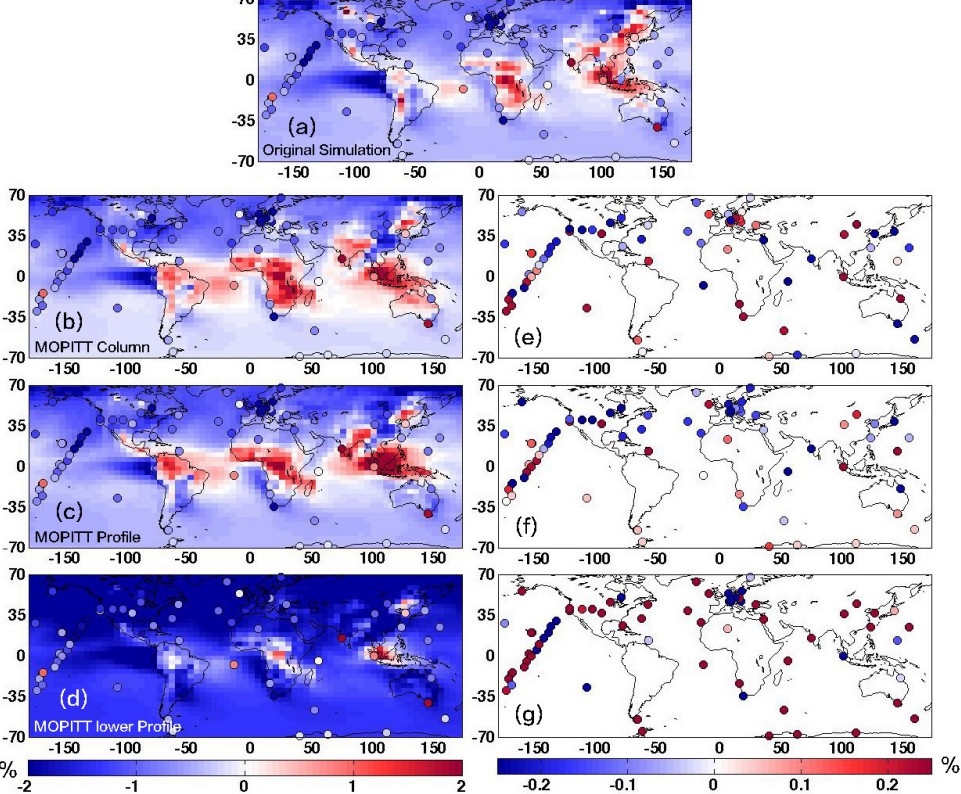


**Figure 9.** Panels (a-d): long-term trend (annual) of surface CO concentration for 2001 – 2015 from WDCGG sites, and model simulations driven with a priori and a posteriori emissions. Panels (e-g): effect of a posteriori emissions, derived by abs($\text{Trend}_{\text{aposteriori}} - \text{Trend}_{\text{WDCGG}}$) - abs($\text{Trend}_{\text{apriori}} - \text{Trend}_{\text{WDCGG}}$); blue (red) means the a posteriori emissions improves (degrades) the agreement with WDCGG measurements compared to the a priori emissions, while white indicates no change from the priori. Only stations with more than 10 year observations (the time range between the first and last observations) during 2001-2015 are included.





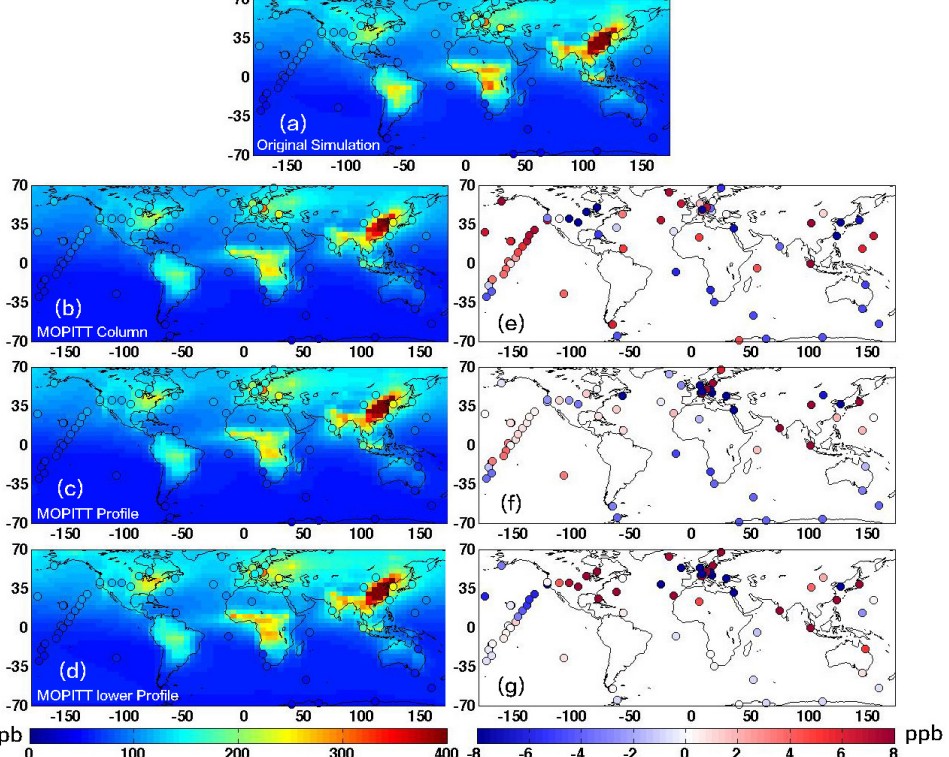


**Figure 10.** Panels (a-d): long-term mean value of surface CO concentration for 2001 – 2015 from WDCGG sites, and model simulations driven with a priori and a posteriori emissions. Panels (e-g): effect of a posteriori emissions, derived by abs($CO_{aposteriori} − CO_{WDCGG}$) - abs($CO_{apriori}$ - $CO_{WDCGG}$); blue (red) means the a posteriori emissions improves (degrades) the agreement with WDCGG measurements compared to the a priori emissions, while white indicates no change from the priori. Only stations with more than 10 year observations (the time range between the first and last observations) during 2001-2015 are included.







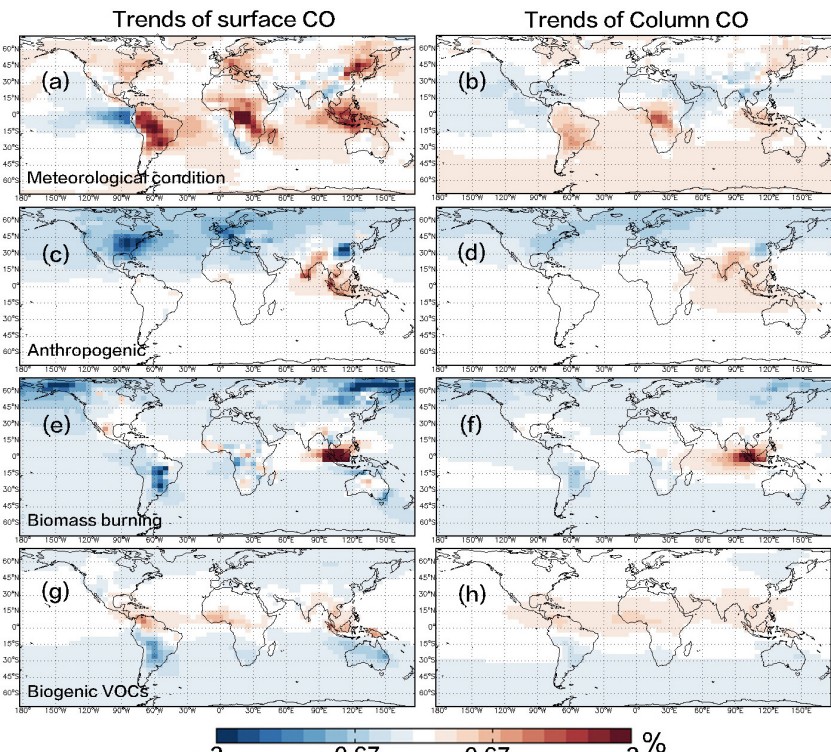


**Figure 11.** Long-term trend (annual) of modeled surface and column CO for 2001 – 2015 with
(a-b) all emission sources are fixed at 2001 level. (c-d) variable anthropogenic emissions; (e-f)
variable biomass burning emissions; (g-h) variable biogenic VOCs emissions; The variable
emissions are constrained with MOPITT profile data.