# Peer review of "A fifteen year record of CO emissions constrained by MOPITT CO"

_Atmospheric Chemistry and Physics, 2016_

## Referee Comment (RC1) · Anonymous Referee #1 · 14 Dec 2016

The authors make an interesting contribution to the quantification of CO surface emissions and of their trend over the past 15 years. I recommend its publication provided the following issues are addressed. Most of them are minor, but a couple of them deserve much more attention.

- l. 80: The authors anticipate on their results, which is not really appropriate in an introduction (it breaks the logic flow).

- l. 97: measurement and model systematic errors can be damped but not suppressed.

- l. 98: "systematic biases" -> "systematic errors".

- l. 143: the previous example of the SCIAMACHY bias is time-dependent. The

authors should explain why they think that the MOPITT bias does not vary much with time (mostly with the season).

- l. 185: the authors seem to neglect the error statistics provided by the retrieval product. We can understand that they prefer raising them at 20% to be conservative, given likely systematic errors, but ignoring the vertical correlations is really surprising. **This point is important because it bears most of the credibility of the following profile/lower profile inversion results vs. column inversion results**. In addition, the ad-hoc uncorrelated observation budget used here is not internally consistent: when summing the profile level (error) covariances, one does not get the column (error) variance. **This inconsistency basically suppresses the possibility to compare the two types of results meaningfully**. Last, model errors are very likely correlated in the vertical and even uncertain large or medium vertical correlations (let us say 0.5 for instance) for this term of the observation error budget are better than the null correlations assumed here.

- l. 186: the authors seem to combine combustion and VOC sources of CO together but later in Section 4.2 they show result by source type. They should explain how they split the information on the source type with simple column or profile retrievals of CO. In particular, I cannot see how VOC sources and their trends can be separated from the rest.

- l. 215-216: This sentence ("... indicated that regional inversions have more advantages than global inversions ... better controlled") is unnecessarily polemical and may actually be wrong depending on how we understand "better controlled". There are pros and cons and the statement cannot leave the impression that the case has been closed.

- l. 219: " model" -> " models" .

- l. 228: the authors need to be clear that they do not use the same land data in the

first and in the second step. Otherwise they would correlate boundary condition errors and observation errors in the second step and possibly induce weird side effects on their results (because those correlations are not accounted for).

- l. 254: Montzka et al. (2011) is recalled, but these authors wrote "Despite the much lower atmospheric CH3CCl3 mixing ratios in recent years (≃13 ppt in 2007), they remained precisely measured through 2007. Precision for the analysis of CH3CCl3 (0.5 to 0.75% as repeatability) has remained comparable to the nearly constant (on a relative basis) standard deviation of paired flask means collected within a month at remote stations of 0.7−1.1% through 2007. Data after the end of 2007 are not included in this report owing to instrumental problems that developed in 2008." The present authors should give the same level of detail and clarify the fact that the instrumental problem does not affect their results.

- l. 276: "demonstrate" is too strong.

- l. 296: there is also an initial increase in the measurements that should be commented.

- l. 313: this is only true for the profile results.

- l. 335: large PBL height errors happen everywhere over the globe. Why should they just affect India and SE Asia ?

- l. 374: these 2014 and 2015 studies are not "more recent" than Field et al. (2016). Actually, the authors could discuss the "more recent" study of Yin et al. (2016) that seems to well overlap with their approach.

- l. 376: extra comma.

- l. 396: the above-mentioned issue in the observation error statistics is also a likely explanation.

- l. 464: to be fair and consistent with the second part of the sentence, the authors should also speak of an update about this question, since it has been (imperfectly) addressed before.

- References should be ordered.

**Reference**

Yin, Y., et al. (2016), Variability of fire carbon emissions in equatorial Asia and its nonlinear sensitivity to El Niño, Geophys. Res. Lett., 43, 10,472–10,479, doi:10.1002/2016GL070971.

---

## Referee Comment (RC2) · Anonymous Referee #2 · 6 Jan 2017

**Review: A fifteen year record of CO emissions constrained by MOPITT CO observations, Z. Jiang et al., 2016, doi:10.5194/acp-2016-811**

The work investigates the possible cause of the observed trend of a reduction of Carbon Mononixde (CO) emissions over the last 15 years over the northern hemisphere and parts of China. This trend is somewhat mitigated by an increased trend of CO emissions over India. The authors use global MOPITT remote sensing data of CO in the thermal infrared region to constrain model forecasts of CO concentrations and surface emissions. The model being used is the adjoint of the off-line global chemistry transport model GEOS-Chem.

The authors make 4 big assumptions: 1) Unknown model biases can be handled with by providing independent boundary conditions of CO concentrations over oceans each month from a Kalman Filter inversion run, 2) Local continental scale emissions can be estimated then by a 4dvar method constrained by MOPITT observations over land (and constrained by the boundary conditions of CO concentrations over the oceans), 3) The inversion system works best by removing a latitudional bias in MOPITT retrievals as derived from the HIAPER Pole to Pole Observations campaign (HIPPO), 4) The hydroxyl radical (OH) variability cannot explain the decrease in CO emissions if we put trust in the method of using MCF (methyl chloroform) measurements as a proxy for estimating atmospheric OH concentration change.

**Comments:**

**- Chapter: 2.1 MOPITT**

Did you do any data thinning on the MOPITT data and how did you screen the MOPITT data?

**- line 176-178:**

You need to describe the 4dvar adjoint method in more detail. What are typical numbers of N and it is not clear from the equation (line 178) or Figure 3 how you defined the length of the assimilation window in your 4dvar system. In GEOS-Chem met fields are typically updated every 6 hours – does this also correspond to your assimilation window (e.g. 6 hour window)? Or is your assimilation window a full month and observations are sampled every hour?

**- line 186-189:**

Cite: D.B.Jones, et al: Potential of observations from the Tropospheric Emission Spectrometer to constrain continental sources of carbon monoxide, doi:10.1029/2003JD003702, J. Geopys. Res, 2003

It is not clear to me why the authors cannot follow the method of constructing the observation error covariance matrix as outlined in the above paper (Dylan et al 2003). Of course TES and MOPITT are different products but as far as I remember MOPITT will also let you construct a retrieval error matrix as part of their released data products (they come with the averaging kernels). It is true that there is some vertical correlation in the averaging kernels but cannot account for the information loss of a uniform or flat construed observation error.

**- 190-196**

Reword and emphasise that posterior emissions estimates (e.g. Table 1) do not have uncertainty reduction error bars because of the way the adjoint method works and ask Daven Henze if there is a reference for that.

**- line 194-196: As shown by Heald et al (2004), different assumptions about the inversion configuration can produce differences in the source estimates that are significantly larger than the a posteriori errors.**

Is this statement related to the bias correction in the next paragraph (line 197-209)? Why is this important here?

**- line 197-198: Removing the bias in initial conditions is essential for inverse analysis, and can be performed with various data assimilation techniques.**

Have you got a reference for this? I have heard people claiming (I am not one of them) that in a good inversion system there is no bias correction needed. Have you tested your system without bias correction?

**- line 218-220: They demonstrated that the systematic bias associated with North American CO emissions due to OH distribution can be reduced by up to 50% with optimised boundary conditions. Similar optimisation on the boundary condition can also be employed in global model, for example, Pfister et al. (2005) constrained biomass burning CO emissions from boreal North America with optimised CO fields outside the impacted region.**

How does this relate to your work? Your are using pre-calculated OH fields from a full chemistry run. Are you making the point here that the influence of the badly understood OH bias can be reduced by optimised CO 3D boundary conditions (e.g. from your Kalman Filter at the beginning of each month)? Please clarify.

**- Figure 3**

This needs clarification in the Figure caption or text. If I am right to assume that your Kalman filter runs from 1st of March until 31nd December first and is

completely independet of the 4dvar inversion in the assimilation window? And there is no feedback of the 4dvar inversion results to the boundary conditions of the following months?

**- 4.1 Long-term variation of global tropospheric OH**

Krol et al. found a somewhat different result of OH trends based on MCF measurements and model studies. Admittedly for a different study period (1978-1998). You could (or should) cite that paper: M. Krol et al., 1998: Global OH trend inferred from methylchloroform measurements, 103, p.10,697—10,711, 1998, J. Geopys. Res.

**- 4.2 Long-term variation of global CO emissions**

It would be a good idea if you split the section into different smaller subsections:

4.2.1 Emissions US
4.2.2 Emissions EU
4.2.3 Emissions India + South East Asia
4.2.4 Biomas Burning Emissions

etc.

**- line 425-427: In a recent study, Schnell et al. 92015) evaluate surface O3 concentrations simulated by multi-models for North America and Europe. They found most models can provide good simulations for the patterns of O3 but cannot reproduce the magnitude.**

I do not think citing an ozone study supports your argument in terms of CO.

**- line 466-468**

Reformulate the part including 'MCF'. I do not think you have used MCF to 'evaluate changes in the sources and sinks of atmospheric CO … '.

**- Table 1**

Add a fifth column of global total posterior emissions to the 3 individual sub tables: 'MOPITT Columns (Tg/year)', 'MOPITT Profile (Tg/year)' and 'MOPITT Lower Profile (Tg/year)'.

Add a sixth column to the 3 individual sub tables for posterior CH4 and VOC production.

Also append 4 single columns for the global prior emissions in each year.

e.g.

Year,US,EU,China,India,CH4,VOC, US,EU,China,India,CH4,VOC, US,EU,China,India,CH4,VOC, PRIOR ANTHRO, PRIOR CH4, PRIOR VOC, PRIOR TOTAL

And comment on these global budgets in the main text.

**- Figure 11**

I am not convinced your method of singling out the meteorological effects works as intented. Firstly, what exactly is being defined as meteorological conditions? I think the accumulation of surface CO, especially over the tropical regions and to a lesser extended the slight loss of CO at higher latitudes is an artifact and CO builds up, unrealistically, in GEOS-Chem tagged tracer mode.

I am not asking you to conduct more model calculations. However, it would have beend interesting to see if a full global 4x5 GEOS-Chem CO chemistry run gives a similar answer than Figure 11a and 11b.

---

## Author Comment (AC1) · 17 Jan 2017

The authors make an interesting contribution to the quantification of CO surface emissions and of their trend over the past 15 years. I recommend its publication provided the following issues are addressed. Most of them are minor, but a couple of them deserve much more attention.

Thank you for your comments. Modifications have been made to improve this manuscript.

Q1: I. 80: The authors anticipate on their results, which is not really appropriate in an introduction (it breaks the logic flow).

Changed.

Q2: I. 97: measurement and model systematic errors can be damped but not suppressed.

Changed.

Q3: I. 98: "systematic biases" -> "systematic errors".

Changed.

Q4: I. 143: the previous example of the SCIAMACHY bias is time-dependent. The authors should explain why they think that the MOPITT bias does not vary much with time (mostly with the season).

The limited measurements provided by the HIPPO aircraft will result in uncertainties in the correction factors, which is more significant in the seasonal average than annual average. On the other hand, we are focusing on the interannual variation of CO emissions. The seasonal variation of CO emissions is not very important in this work. Consequently, we decided to use the annual mean correction factor. More description has been added.

Q5: I. 185: the authors seem to neglect the error statistics provided by the retrieval product. We can understand that they prefer raising them at 20% to be conservative, given likely systematic errors, but ignoring the vertical correlations is really surprising. This point is important because it bears most of the credibility of the following profile/lower profile inversion results vs. column inversion results. In addition, the ad-hoc uncorrelated observation budget used here is not internally consistent: when summing the profile level (error) covariances, one does not get the column (error) variance. This inconsistency basically suppresses the possibility to compare the two types of results meaningfully. Last, model errors are very likely correlated in the vertical and even uncertain large or medium vertical correlations (let us say 0.5 for instance) for this term of the observation error budget are better than the null correlations assumed here.

A very good question! We have compared the discrepancies associated with two types
of error covariance matrix in the preparation stage of this work: 1) diagonal matrix (this work); 2) full error covariance matrix including vertical correlation, based on MOPITT error covariance. Our results show that the difference in the scaling factors is small, perhaps due to the large amount of satellite measurements in our global scale inversion. Because we are focusing on the mitigation of effects of systematic errors, we used the diagonal matrix to keep consistency with our previous studies. However, as the reviewer indicated, a better description for the error covariance matrix is important. We will improve our methodology in our future study.

Q6: I. 186: the authors seem to combine combustion and VOC sources of CO together but later in Section 4.2 they show result by source type. They should explain how they split the information on the source type with simple column or profile retrievals of CO. In particular, I cannot see how VOC sources and their trends can be separated from the rest.

As the reviewer indicated, we cannot completely separate the a posteriori emission estimates from different sources. However, the various spatial and temporal distribution of emissions sources (e.g. anthropogenic vs. biomass burning) provides valuable information to distinguish the contribution from each category. In order to further isolate the influences of biomass burning, the months dominated by biomass burning (biomass burning CO > 50% of total CO emission in an individual grid) are excluded in the trend analysis for anthropogenic and VOC sources (Figure 5). More description has been added.

Q7: I. 215-216: This sentence (": : : indicated that regional inversions have more advantages than global inversions : : : better controlled") is unnecessarily polemical and may actually be wrong depending on how we understand "better controlled". There are pros and cons and the statement cannot leave the impression that the case has been closed.

Thank you for your suggestion! The statement has been changed.
Q8: I. 219: "model" -> "models" .

Changed.

Q9: I. 228: the authors need to be clear that they do not use the same land data in the first and in the second step. Otherwise they would correlate boundary condition errors and observation errors in the second step and possibly induce weird side effects on their results (because those correlations are not accounted for).

I am sorry for the confusion. In the two-step approach:

Step 1: We directly modify CO concentrations using sequential Kalman filter assimilation. Both MOPITT data over land and ocean are used. Step 2: We constrain CO emissions over land with MOPITT data over land only. The boundary condition is from step 1.

The objective of Step 1 is to provide the best global CO fields, based on MOPITT. We need to assimilate MOPITT data over land in the first step to keep the consistency between boundary conditions and emissions.

Q10: I. 254: Montzka et al. (2011) is recalled, but these authors wrote "Despite the much lower atmospheric CH3CCI3 mixing ratios in recent years ('13 ppt in 2007), they remained precisely measured through 2007. Precision for the analysis of CH3CCI3 (0.5 to 0.75% as repeatability) has remained comparable to the nearly constant (on a relative basis) standard deviation of paired flask means collected within a month at remote stations of 0.7ôĂĂĂ1.1% through 2007. Data after the end of 2007 are not included in this report owing to instrumental problems that developed in 2008." The present authors should give the same level of detail and clarify the fact that the instrumental problem does not affect their results.

The website (NOAA) shows: "NOAA flask data obtained by the GCMS for some compounds analyzed during the 2008.5-2009.5 period are subject to some small biases owing to instrumental issues during that period. Data obtained for CH3CCI3 during
that time period, for example, should not be used for deriving hydroxyl radical concentrations"

According to Figure 4, we believe the influence of the instrumental problems (2008.5-2009.5) on our analysis (2001-2015) is small.

Q11: I. 276: "demonstrate" is too strong.

Changed.

Q12: I. 296: there is also an initial increase in the measurements that should be commented.

The initial increase at 2001-2002 could be caused by uncertainties in the data. We are trying to avoid to make a conclusion about trend based on short (2 years) period data. A sentence has been added for this issue.

Q13: I. 313: this is only true for the profile results.

As shown in Table 1, an increase of Chinese emissions from 2001 to 2004 is shown by all three analyses.

Q14: I. 335: large PBL height errors happen everywhere over the globe. Why should they just affect India and SE Asia?

Thank you for pointing out this issue. We have removed "PBL height" in the discussion.

Q15: I. 374: these 2014 and 2015 studies are not "more recent" than Field et al.(2016). Actually, the authors could discuss the "more recent" study of Yin et al. (2016) that seems to well overlap with their approach.

The discussion has been changed. We didn't cite Yin's work here, because we hope to demonstrate the consistency between our inversion results with studies using different approach (not an inverse modelling).

Q16: I. 376: extra comma.
Changed.

Q17: I. 396: the above-mentioned issue in the observation error statistics is also a likely explanation.

The lower tropospheric profile data includes the lowest three levels (1000hPa, 900hPa and 800 hPa). The influence of correlation of these three levels should be small.

Q18: I. 464: to be fair and consistent with the second part of the sentence, the authors should also speak of an update about this question, since it has been (imperfectly) addressed before.

Changed.

Q19: References should be ordered.

Changed.

---

## Author Comment (AC2) · 17 Jan 2017

The work investigates the possible cause of the observed trend of a reduction of Carbon Mononixde (CO) emissions over the last 15 years over the northern hemisphere and parts of China. This trend is somewhat mitigated by an increased trend of CO emissions over India. The authors use global MOPITT remote sensing data of CO in the thermal infrared region to constrain model forecasts of CO concentrations and surface emissions. The model being used is the adjoint of the off-line global chemistry transport model GEOS-Chem.

The authors make 4 big assumptions: 1) Unknown model biases can be handled with by providing independent boundary conditions of CO concentrations over oceans each month from a Kalman Filter inversion run, 2) Local continental scale emissions can

be estimated then by a 4dvar method constrained by MOPITT observations over land (and constrained by the boundary conditions of CO concentrations over the oceans), 3) The inversion system works best by removing a latitudinal bias in MOPITT retrievals as derived from the HIAPER Pole to Pole Observations campaign (HIPPO), 4) The hydroxyl radical (OH) variability cannot explain the decrease in CO emissions if we put trust in the method of using MCF (methyl chloroform) measurements as a proxy for estimating atmospheric OH concentration change.

Thank you for your comments. Modifications have been made to improve this manuscript.

Q1: Chapter: 2.1 MOPITT: Did you do any data thinning on the MOPITT data and how did you screen the MOPITT data?

We employed the same data quality control as our previous studies. Detailed description has been added in Section 2.1.

Q2: line 176-178: You need to describe the 4dvar adjoint method in more detail. What are typical numbers of N and it is not clear from the equation (line 178) or Figure 3 how you defined the length of the assimilation window in your 4dvar system. In GEOS-Chem met fields are typically updated every 6 hours – does this also correspond to your assimilation window (e.g. 6 hour window)? Or is your assimilation window a full month and observations are sampled every hour?

Thank you for your suggestion! More description has been added.

In order to match model output, the high resolution MOPITT measurements are averaged temporally (one-hour resolution) and spatially ($4°$x$5°$ resolution) to produce grid mean observations. The length of assimilation window is one month. The number (N) of grid mean observations in one month is around 10000.

Q3: line 186-189: Cite: D.B.Jones, et al: Potential of observations from the Tropospheric Emission Spectrometer to constrain continental sources of carbon monoxide,

doi:10.1029/2003JD003702, J. Geopys. Res, 2003
It is not clear to me why the authors cannot follow the method of constructing the observation error covariance matrix as outlined in the above paper (Dylan et al 2003). Of course TES and MOPITT are different products but as far as I remember MOPITT will also let you construct a retrieval error matrix as part of their released data products (they come with the averaging kernels). It is true that there is some vertical correlation in the averaging kernels but cannot account for the information loss of a uniform or flat construed observation error.

Jones et al. (2003) used the NMC method to assess the model transport errors. This approach uses pairs of model forecasts, of different length, but which are valid for the same time, to characterize the model errors. We do not have such forecasts available during this analysis period.

We have compared the discrepancies associated with two types of error covariance matrix in the preparation stage of this work: 1) diagonal matrix (this work); 2) full error covariance matrix including vertical correlation, based on MOPITT error covariance. Our results show that the difference in the scaling factors is small, perhaps due to the large amount of satellite measurements in our global scale inversion.

Because we are focusing on the mitigation of effects of systematic errors, we used the diagonal matrix to keep consistency with our previous studies. However, as the reviewer indicated, a better description for the error covariance matrix is important. We will improve our methodology in our future study.

Q4: 190-196: Reword and emphasise that posterior emissions estimates (e.g. Table 1) do not have uncertainty reduction error bars because of the way the adjoint method works and ask Daven Henze if there is a reference for that.

Thank you for your suggestion! The discussion has been modified.

Q5: line 194-196: "As shown by Heald et al (2004), different assumptions about the

inversion configuration can produce differences in the source estimates that are significantly larger than the a posteriori errors." Is this statement related to the bias correction in the next paragraph (line 197-209)? Why is this important here?

We hope to demonstrate that the actual a posteriori uncertainty (including systematic errors) is much larger than the a posteriori uncertainty calculated based on Gaussian assumption (random errors).

Q6: line 197-198: "Removing the bias in initial conditions is essential for inverse analysis, and can be performed with various data assimilation techniques." Have you got a reference for this? I have heard people claiming (I am not one of them) that in a good inversion system there is no bias correction needed. Have you tested your system without bias correction?

We have tested the effects of initial condition in our previous study. As shown in Figure 4a of Jiang et al. (2013), there are large discrepancies between MOPITT and original model simulation due to the accumulation of model errors prior to the assimilation window. We cannot use the biased initial condition for the inverse analysis.

"a good inversion system there is no bias correction needed" is valid for the ideal condition. However, there are always systematic biases, and we cannot ignore them. For example, Figure 1 shows noticeable discrepancies between MOPITT and HIPPO. We have to mitigate these discrepancies using latitude dependent correction factors, although we know the best approach is an update of retrieval algorithm.

Q7: line 218-220: "They demonstrated that the systematic bias associated with North American CO emissions due to OH distribution can be reduced by up to 50% with optimised boundary conditions. Similar optimisation on the boundary condition can also be employed in global model, for example, Pfister et al. (2005) constrained biomass burning CO emissions from boreal North America with optimised CO fields outside the impacted region."

How does this relate to your work? Your are using pre-calculated OH fields from a full chemistry run. Are you making the point here that the influence of the badly understood OH bias can be reduced by optimised CO 3D boundary conditions (e.g. from your Kalman Filter at the beginning of each month)? Please clarify.

We hope to demonstrate that the influences of systematic errors can be mitigated by the optimization on the boundary condition. We have changed the statement to make it more concise.

As the reviewer indicated, the optimization on the boundary conditions (e.g. around North America) can really mitigate the influences of OH bias on a posteriori estimation of North American CO emissions. Although the OH distribution over North America continent is still biased in a reginal inversion, the adverse effects of biased OH distribution on the CO inflow from outside of North America can be significantly reduced.

Q8: Figure 3: This needs clarification in the Figure caption or text. If I am right to assume that your Kalman filter runs from 1st of March until 31nd December first and is completely indepednet of the 4dvar inversion in the assimilation window? And there is no feedback of the 4dvar inversion results to the boundary conditions of the following months?

Thank you for your suggestion! The Figure caption has been changed.

Q9: 4.1 Long-term variation of global tropospheric OH. Krol et al. found a somewhat different result of OH trends based on MCF measurements and model studies. Admittedly for a different study period (1978-1998). You could (or should) cite that paper: M. Krol et al., 1998: Global OH trend inferred from methylchloroform measurements, 103, p.10,697—10,711, 1998, J. Geopys. Res.

The citation has been added.

Q10: 4.2 Long-term variation of global CO emissions. It would be a good idea if you split the section into different smaller subsections: 4.2.1 Emission;s US 4.2.2 Emissions EU; 4.2.3 Emissions India + South East Asia; 4.2.4 Biomas Burning Emissions etc.

Thank you for your suggestion! Two subsections "Regional analysis for anthropogenic emissions" and "Regional analysis for biomass burning emissions" have been added.

Q11: line 425-427: "In a recent study, Schnell et al. 92015) evaluate surface O3 concentrations simulated by multi-models for North America and Europe. They found most models can provide good simulations for the patterns of O3 but cannot reproduce the magnitude." I do not think citing an ozone study supports your argument in terms of CO.

This citation has been removed.

Q12: line 466-468. Reformulate the part including 'MCF'. I do not think you have used MCF to 'evaluate changes in the sources and sinks of atmospheric CO . . . '.

The statement has been modified.

Q13: Table 1. Add a fifth column of global total posterior emissions to the 3 individual sub tables: 'MOPITT Columns (Tg/year)', 'MOPITT Profile (Tg/year)' and 'MOPITT Lower Profile (Tg/year)'.

Add a sixth column to the 3 individual sub tables for posterior CH4 and VOC production. Also append 4 single columns for the global prior emissions in each year. e.g. Year,US,EU,China,India,CH4,VOC,  US,EU,China,India,CH4,VOC,US,EU,China,India CH4,VOC, PRIOR ANTHRO, PRIOR CH4, PRIOR VOC, PRIOR, TOTAL, And comment on these global budgets in the main text.

Thank you for your suggestion! Three columns of global total anthropogenic emissions have been added in Table 1. A new table (Table 2) was added to show the annual variation of biomass burning emissions.

We didn't provide values for CO sources from VOC and CH4 oxidization because our

results for these two sources are inconclusive. The values for a priori emissions are also excluded because the tables are already complex.

Q14: Figure 11. I am not convinced your method of singling out the meteorological effects works as intented. Firstly, what exactly is being defined as meteorological conditions? I think the accumulation of surface CO, especially over the tropical regions and to a lesser extened the slight loss of CO at higher latitudes is an artifact and CO builds up, unrealistically, in GEOS-Chem tagged tracer mode. I am not asking you to conduct more model calculations. However, it would have beend interesting to see if a full global 4x5 GEOS-Chem CO chemistry run gives a similar answer than Figure 11a and 11b.

A very good question! Our forward model simulation, based on various versions of the meteorological fields (i.e. GEOS-4, GEOS-5 and GEOS-FP), is not an ideal tool for the analysis of influences of meteorological fields. We have modified the text to emphasize on this point.